# Significance of Circulating Remnant Lipoprotein Cholesterol Levels Measured by Homogeneous Assay in Patients with Type 2 Diabetes

**DOI:** 10.3390/biom13030468

**Published:** 2023-03-03

**Authors:** Kazumi Matsushima-Nagata, Takeshi Matsumura, Yuki Kondo, Kensaku Anraku, Kazuki Fukuda, Mikihiro Yamanaka, Masahiro Manabe, Tetsumi Irie, Eiichi Araki, Hiroyuki Sugiuchi

**Affiliations:** 1Department of Medical Technology, Kumamoto Health Science University, Kumamoto 861-5598, Japan; matusima@kumamoto-hsu.ac.jp (K.M.-N.); anraku@kumamoto-hsu.ac.jp (K.A.); sugiuchi@kumamoto-hsu.ac.jp (H.S.); 2Department of Metabolic Medicine, Faculty of Life Sciences, Kumamoto University, Kumamoto 860-8556, Japan; fukudakazuki@hotmail.com (K.F.); earaki@gpo.kumamoto-u.ac.jp (E.A.); 3Department of Clinical Chemistry and Informatics, Graduate School of Pharmaceutical Sciences, Kumamoto University, Kumamoto 862-0973, Japan; ykondo@kumamoto-u.ac.jp (Y.K.); tirie@gpo.kumamoto-u.ac.jp (T.I.); 4Research and Development/Technology Research Laboratory, Shimadzu Corporation, Kyoto 604-8511, Japan; yamanaka.mikihiro.2uc@shimadzu.co.jp; 5Department of Laboratory Medicine, Kumamoto University Hospital, Kumamoto 860-8556, Japan; masahiro-manabe@kuh.kumamoto-u.ac.jp

**Keywords:** remnant lipoprotein, type 2 diabetes, remnant lipoprotein cholesterol homogeneous assay, cardiovascular disease

## Abstract

Remnant lipoproteins (RLs), which are typically present at high concentrations in patients with type 2 diabetes mellitus (T2DM), are associated with cardiovascular disease (CVD). Although an RL cholesterol homogeneous assay (RemL-C) is available for the measurement of RL concentrations, there have been no studies of the relationship between RemL-C and clinical parameters in T2DM. Therefore, we evaluated the relationships between RemL-C and CVD-related parameters in patients with T2DM. We performed a cross-sectional study of 169 patients with T2DM who were hospitalized at Kumamoto University Hospital. Compared with those with low RemL-C, those with higher RemL-C had higher fasting plasma glucose, homeostasis model assessment for insulin resistance (HOMA-R), total cholesterol, triglyceride, small dense low-density lipoprotein cholesterol (sdLDL-C), and urinary albumin-creatinine ratio; and lower high-density lipoprotein cholesterol, adiponectin, and ankle brachial pressure index (ABI). Multivariate logistic regression analysis showed that sdLDL-C and ABI were significantly and independently associated with high RemL-C. Although LDL-C was lower in participants with CVD, there was no difference in RemL-C between participants with or without CVD. Thus, RemL-C may represent a useful index of lipid and glucose metabolism, and that may be a marker of peripheral atherosclerotic disease (PAD) in male patients with T2DM.

## 1. Introduction

Remnant lipoproteins, which are present at high circulating concentrations when lipoprotein metabolism is impaired, are associated with the progression of atherosclerosis and coronary artery disease (CAD) [1,2]. It is well known that high serum triglyceride (TG) concentrations are an important risk factor for coronary heart disease [3,4,5]. However, it has been reported that the TG-rich remnant lipoprotein cholesterol concentration is closely associated with CAD risk in patients with normal TG concentrations [6]. Therefore, remnant lipoprotein cholesterol concentration may represent a more sensitive marker of CAD risk than TG.

High serum TG concentrations are frequently present in patients with diabetes and are thought to be important contributors to the onset of cardiovascular disease (CVD) in patients with diabetes. Indeed, the Framingham Heart Study revealed that diabetes is an independent risk factor for CAD: its presence results in a 1.5-fold higher multivariate-adjusted risk of CAD in men and a 1.8-fold higher adjusted risk in women [7]. The Hisayama Study also demonstrated that type 2 diabetes mellitus (T2DM) significantly increases the risks of both cerebral infarction and CAD during a five-year follow-up period in a general Japanese population [8]. In addition, the serum TG concentration is a key predictor of CAD, having comparable value to low-density lipoprotein (LDL)-cholesterol (LDL-C) and glycated hemoglobin (HbA1c) in Japanese patients with T2DM [9]. This suggests that remnant lipoproteins, which contain a large amount of TG, may also be a predictor of CAD in T2DM.

During the course of atherosclerosis, cerebrovascular disease and peripheral arterial disease are observed, as well as the development of ischemic heart disease. [10]. In the case of atherosclerosis of the arteries of the lower extremities, this typically develops in multiple stages; the arteries distal to the knee are more commonly affected; and it is characterized by the presence of calcification of the middle layers of the arterial walls in patients with diabetes. Invasive revascularization therapy, including percutaneous balloon angioplasty with optional stent placement, is important for patients with CAD, but the phenomenon of restenosis, which is especially common in patients with diabetes, limits the long-term effectiveness of this treatment [11].

One of the clinically available methods to measure the cholesterol concentration associated with remnant lipoproteins is a remnant-like particle (RLP)-cholesterol (RLP-C) assay, which is an immunoaffinity separation method (RLP-C assay; Otsuka, Tokyo, Japan). This assay involves the isolation of RLPs from human serum using an immunoaffinity gel containing two different immobilized monoclonal antibodies against human apolipoprotein (Apo)-AI and Apo-B-100 [12,13]. Many clinical studies have shown that RLP-C is a risk factor for CVD, and the serum RLP-C concentration is higher in patients with CAD, diabetes, or metabolic syndrome than in healthy individuals [14,15,16]. Therefore, the RLP-C concentration represents a useful predictor of CVD and has the advantage that it can be measured without the requirement for ultracentrifugation. However, this measurement is relatively time-consuming and an autoanalyzer cannot be used.

After the clinical application of the RLP-C assay, a remnant lipoprotein cholesterol homogeneous assay (RemL-C assay; Kyowa Medex, Tokyo, Japan) has been developed that uses a special surfactant [polyoxyethylene-polyoxybutylene (POE-POB) block copolymer] and phospholipase D, which selectively solubilizes and degrades TG-rich remnant lipoproteins, very low-density lipoprotein (VLDL) remnants, and chylomicron remnants [17,18]. Nakada et al. [18] reported that remnant lipoproteins measured using RemL-C are present at high concentrations in patients with CAD, suggesting that this assay may be useful for coronary risk assessment. However, the relationships between RemL-C and risk factors for CVD have not previously been evaluated. Therefore, in the present study, we aimed to characterize the relationships between RemL-C and several parameters related to CVD in patients with diabetes and to assess the utility of RemL-C measurement for patients with T2DM.

## 2. Materials and Methods

### 2.1. Participants

We performed a cross-sectional study of 212 patients with T2DM who were hospitalized to improve their glycemic control at Kumamoto University Hospital between October 2012 and December 2015. T2DM was diagnosed using the World Health Organization criteria [19]. Patients with type 1 diabetes were excluded, as were those who were positive for glutamic acid decarboxylase (GAD) antibodies, those with a history of ketoacidosis, and those dependent on insulin therapy for their survival. Patients with severe hepatic disease, malignancy, or acute/chronic inflammatory disease were also excluded (Figure 1). Data from a total of 169 patients with T2DM (98 men and 71 women) were analyzed. The characteristics of the participants are listed in Table 1. All the participants were Japanese, and each participated in a detailed interview regarding his/her clinical condition and smoking history.

Hypertension was defined using the administration of any anti-hypertensive medication, a systolic blood pressure (SBP) ≥ 140 mmHg, or a diastolic blood pressure (DBP) ≥ 90 mmHg, according to the criteria of the Japanese Society of Hypertension [20]. Hyperlipidemia was defined as having total cholesterol (TC) > 5.7 mmol/L and/or a TG > 1.7 mmol/L or being under treatment with an antihyperlipidemic agent. CVD was defined as the presence or history of stroke, ischemic heart disease, or arteriosclerosis obliterans.

### 2.2. Compliance with Ethics Guidelines

All the procedures in the study involving human participants were performed in accordance with the ethics standards of the institutional and/or national research committee and the 1964 Helsinki Declaration and its later amendments or comparable ethical standards. The study protocol was approved by the Human Ethics Review Committee of Kumamoto University (Protocol Number: 1468) and registered at UMIN-CTR (UMIN000010436). All the participants provided their written informed consent.

### 2.3. Laboratory Measurements

Blood samples were collected from all the participants in the early morning after an overnight fast. Enzymatic methods (Minaris Medical Co. Ltd., Tokyo, Japan) were used to measure the serum TC and TG concentrations. Homogeneous methods were used to measure the RemL-C, high-density lipoprotein cholesterol (HDL-C) (Minaris Medical Co. Ltd., Tokyo, Japan), and small dense LDL-C (sdLDL) (Denka Seiken Co. Ltd., Tokyo, Japan) concentrations. Immunoturbidimetric methods (Sekisui Medical Co. Ltd., Tokyo, Japan) were used to measure the Apo-AI, Apo-AII, Apo-B, Apo-CIII, and Apo-E concentrations. The enzymatic and immunoturbidimetric measurements were performed on a Hitachi 7180 Auto Analyzer (Hitachi High-Tech Corporation, Tokyo, Japan). Latex aggregation methods (Denka Seiken Co. Ltd., Tokyo, Japan) were used for the measurement of lipoprotein(a) (Lp(a)) and adiponectin concentrations on the Hitachi 7180 Auto Analyzer. An oxygen electrode method (Arkray, Inc., Kyoto, Japan) was used to measure the fasting plasma glucose (FPG) concentration on a GA1170. An ion exchange chromatography method (Tosoh Corporation, Tokyo, Japan) was used to quantify HbA1c on an HLC-732 G8. An electrochemiluminescence method (Roche Diagnostics K.K., Tokyo, Japan) was used to measure fasting plasma insulin (FPI) concentration on a Modular Analytics E module. A latex aggregation method (Denka Seiken Co. Ltd., Tokyo, Japan) was used to measure high-sensitivity C-reactive protein (hsCRP) concentration on a BM2250. The measurement of the interleukin-6 (IL-6) and tumor necrosis factor-α (TNF-α) concentrations was outsourced to SRL, Inc. (Tokyo, Japan). LDL-C concentration was measured using the Friedewald formula [21]. HbA1c (%) was calculated as National Glycohemoglobin Standardization Program (NGSP) equivalent values (%) using the formula HbA1c (%) = 1.02 × HbA1c [Japan Diabetes Society (JDS)] (%) + 0.25%, considering the HbA1c (JDS) (%) levels measured relative to the previous Japanese standard substance, the measurement method, and the HbA1c (NGSP) [22]. Estimated glomerular filtration rate (eGFR) was calculated using the age, sex, and serum creatinine concentration and the equation for Japanese people of Matsuo et al. [23]. The urinary albumin-creatinine ratio (ACR) was calculated using the urinary albumin and creatinine concentrations measured in early-morning fasting spot urine samples. Insulin resistance was assessed using the homeostasis model assessment for insulin resistance (HOMA-R), calculated using the fasting concentrations of insulin and glucose using the following formula: HOMA-R = FPI (μU/mL) × FPG (mmol/L)/22.5 [24].

### 2.4. Carotid Ultrasonography and the Measurement of Ankle Brachial Pressure Index (ABI) and Brachial-Ankle Pulse Wave Velocity (baPWV)

The intima-media thickness (IMT) of the common carotid artery (CCA-IMT) was measured using an ultrasound machine (Shimadzu SDU-2200, Shimadzu Co., Ltd., Kyoto, Japan) and a transducer frequency of 5–10 MHz, which provides an axial resolution of 0.30 mm. Intima Scope software (Media Cross Co. Ltd., Tokyo, Japan) was used for computer-assisted acquisition, processing, the storage of B-mode images, and the calculation of CCA-IMT. CCA-IMT was defined and measured as previously described [25]. The mean CCA-IMT and the maximum value of CCA-IMT (max CCA-IMT) were recorded.

ABI and baPWV were measured automatically using an ABI-form (BP-203RPE II; Nippon Colin, Komaki, Japan). The lower value of ABI and the higher values of baPWV recorded for the left and the right sides were used in data analysis.

### 2.5. Statistical Analyses

All the statistical analyses were performed using JMP software version 13.0 (SAS Institute, Cary, NC, USA). Values are presented as mean ± SEM or as numbers. Categorical data were compared using the chi-square test. Multivariate analysis of the relationships of low RemL-C (<0.24 mmol/L) and high RemL-C (≥0.24 mmol/L) with other parameters was performed using logistic regression. The relationships between RemL-C as a continuous variable and other parameters were evaluated using Pearson’s correlation coefficients, and in the multivariate analysis, stepwise linear regression was performed. Because some of the variables showed skewed distributions (duration of diabetes, HOMA-R, TG, hs-CRP, RemL-C, Lp(a), adiponectin, TNF-α, IL-6, eGFR, and ACR), these data were logarithmically transformed before this analysis. *p* values < 0.05 were considered to be statistically significant.

## 3. Results

### 3.1. Characteristics of the Participants

The characteristics of the participants as a whole and of those with lower or higher RemL-C concentrations are shown in Table 1. Compared with participants with low RemL-C, those with higher RemL-C had higher TC, TG, and sdLDL-C and lower HDL-C (Table 1). Besides lipid metabolism, higher RemL-C had higher FPG and HOMA-R and lower adiponectin, suggesting that RemL-C was affected by obese-related glucose metabolism (Table 1). Interestingly, higher RemL-C had higher ACR and lower ABI, suggesting the relevance between RemL-C and diabetic nephropathy and/or peripheral arterial disease in T2DM. Regarding drug therapy, high RemL-C concentrations were present in significantly fewer participants being treated with a thiazolidinedione or a statin (Table 1).

### 3.2. Relationships between RemL-C and Clinical Parameters in the Participants with T2DM

Next, we investigated the relationships between a continuous variable of RemL-C and clinical parameters in the participants with T2DM using Pearson’s correlation coefficients. In relation to lipid metabolism, univariate analysis revealed that RemL-C positively correlated with TC, TG, LDL-C, and sdLDL-C and negatively correlated with HDL-C (Table 2). Regarding glucose metabolism, RemL-C positively correlated with FPG, FPI, HOMA-R, and HbA1c and negatively correlated with adiponectin in participants with T2DM (Table 2).

To clarify whether the RemL-C measurements reflected the serum concentrations of remnant lipoproteins, the correlations between RemL-C and apolipoproteins were evaluated in participants with T2DM. Univariate analysis revealed that RemL-C positively correlated with Apo-B, Apo-CII, Apo-CIII, and Apo-E, which were the major apolipoproteins in remnant lipoprotein particles, and negatively correlated with Apo-AI (Table 3). Multivariate stepwise regression analysis revealed that Apo-AI, Apo-B, Apo-CIII, and Apo-E were significantly and independently associated with RemL-C (Table 3), suggesting that RemL-C measurements reflected the serum concentrations of remnant lipoproteins.

### 3.3. Relationships between High RemL-C and Clinical Parameters in T2DM

Next, to evaluate the independent risk factors associated with high RemL-C, multivariate logistic regression analysis was performed in the participants with T2DM. For multivariate analysis, HOMA-R, sdLDL-C, eGFR, ACR, adiponectin, and ABI were selected as relevant factors from the results of univariate analysis (Table 1). As a result, sdLDL-C and ABI were significantly and independently associated with high RemL-C concentration (Table 4). The inclusion of thiazolidine use and statin use in the multivariate logistic regression analysis had no effect on the results (data not shown).

### 3.4. Effect of Sex on the Relationships between RemL-C and Various Clinical Parameters

Next, to determine the effect of sex, the relationships between RemL-C and various clinical parameters were evaluated separately in participants of each sex. In relation to lipid metabolism, univariate analysis revealed that RemL-C positively correlated with TC, TG, LDL-C, and sdLDL-C and negatively correlated with HDL-C (Table 5). Besides lipid metabolism, higher RemL-C had higher HOMA-R and lower ABI (Table 5). For multivariate logistic regression analysis, HOMA-R, sdLDL-C, age, adiponectin, and ABI were selected as relevant factors from the results of univariate analysis (Table 5). As a result, sdLDL-C and ABI were significantly and independently associated with high RemL-C concentration in the men, as well as in the entire group of participants with T2DM (Table 5). Interestingly, in the female participants, univariate analysis revealed that age, FPG, TG, HDL-C, LDL-C, sdLDL-C, and adiponectin were associated with high RemL-C concentration (Table 6). For multivariate logistic regression analysis, ABI, HOMA-R, sdLDL-C, adiponectin, and age were selected as relevant factors from the results of univariate analysis (Table 6). As a result, sdLDL-C alone was significantly and independently associated with high RemL-C concentration in the women (Table 6).

### 3.5. Differences in Several Parameters between Participants with or without a History of CVD in T2DM

Finally, the differences in parameters in participants with T2DM who did or did not have a history of CVD were evaluated. Age, the duration of T2DM, FPI, Lp(a), mean CCA-IMT, and max CCA-IMT, which were thought to be risk factors for CVD, were significantly higher in participants with CVD than in those without CVD (Table 7). Moreover, eGFR, which was a marker of chronic kidney disease, and ABI, which was a marker of peripheral atherosclerotic disease (PAD), were significantly lower in participants with CVD (Table 7). However, TC and LDL-C, which were strongly risk factors for CVD, were oppositely and significantly lower in participants with CVD (Table 7), suggesting the effect of statin use. Moreover, there were no differences in other risk factors for CVD, such as RemL-C, sdLDL-C, HDL-C, TG, SBP, DBP, BMI, or HOMA-R between these groups (Table 7), suggesting the existence of residual risk for CVD.

## 4. Discussion

To the best of our knowledge, this is the first study to investigate the relationships between RemL-C and clinical parameters related to CVD risk in patients with T2DM. In this observational study, we found that RemL-C is associated with FPG, HOMA-R, TC, TG, HDL-C, LDL-C, sdLDL-C, ACR, adiponectin, and ABI. Logistic regression analysis showed that sdLDL-C and ABI are independently associated with a high RemL-C concentration and that the relationship with ABI is male-specific. However, we found no association between a history of CVD and RemL-C.

There are two clinically available methods for the measurement of remnant lipoprotein cholesterol concentrations. The first, for RLP-C, is an immunoaffinity separation method that is convenient because it does not require ultracentrifugation [12]. The second method, the RemL-C assay, can be performed by homogeneous method [17]. Currently, these two measurements are selected by each investigator. Yoshida et al. previously reported a significant correlation between RemL-C and RLP-C in non-diabetic individuals [26]. In the present study, RemL-C was found to significantly positively correlate with Apo-B, Apo-CII, Apo-CIII, and Apo-E, which are the major apolipoproteins in remnant lipoprotein particles. Therefore, RemL-C likely reflects the circulating remnant lipoprotein particle concentration, even in patients with T2DM.

DM is closely associated with CVD. Conventional cardiometabolic risk factors, such as hypertension, dyslipidemia, and obesity, which are common in T2DM, synergistically increase the risk of CVD. In addition, many previous studies have shown that T2DM is usually characterized by genetic predisposition, hyperglycemia, obesity, and insulin resistance. In particular, obesity and the associated insulin resistance are recognized as potent risk factors for CVD [27]. There is a tendency for hypertriglyceridemia to be present in patients with T2DM, because of the high concentrations of TG-rich lipoproteins, including TG-rich remnant lipoproteins, VLDL remnants, and chylomicron remnants. These high concentrations of TG-rich lipoproteins are thought to increase CVD risk directly and by causing a decrease in HDL and an increase in sdLDL [27]. Consistent with this, in the present study, we found that RemL-C positively correlates with sdLDL-C and negatively correlates with HDL-C. However, whereas RemL-C and RLP-C have previously been reported to be associated with CVD [14,15,16,18], the present findings show no relationship between RemL-C and a history of CVD in patients with T2DM, probably because of therapeutic interventions. Furthermore, LDL-C, a potent risk factor for CVD, showed a significant negative correlation with a history of CVD in the present cohort. However, the lack of a relationship between RemL-C and CVD history, while LDL-C did show an association, may indicate inadequate management of RemL-C. It has been reported that statin therapy significantly reduces the serum RemL-C concentration of hypercholesterolemic patients [28]. Moreover, the Diabetes Atherosclerosis Intervention Study (DAIS) revealed that treatment with fenofibrate reduces serum RLP-C in patients with T2DM [29]. Therefore, it is necessary to raise awareness of the fact that RemL-C represents a residual risk factor for CVD and to implement therapeutic interventions targeting remnant lipoproteins, the efficacy of which can easily be monitored using RemL-C.

Previous studies have shown that remnant lipoproteins are associated with several clinical parameters related to CVD. de Graaf et al. reported that RLP-C positively correlates with age, BMI, SBP, DBP, TC, TG, LDL-C, Apo-B, serum glucose, FPI, and HOMA-R and negatively correlates with HDL-C in patients without combined hyperlipidemia [30]. In addition, Saeed et al. reported that RLP-C is positively associated with TC, TG, LDL-C, and sdLDL-C and negatively associated with HDL-C and Lp(a) in middle-aged adults [31]. In the Framingham Heart offspring study, RLP-C was found to positively correlate with age, BMI, SBP, DBP, serum glucose, TC, TG, and LDL-C and negatively correlate with HDL-C in women [32]. Although there have been no previous equivalent studies using RemL-C, in the present study, we have shown that RemL-C positively correlates with FPG, FPI, HOMA-R, HbA1c, TC, TG, LDL-C, and sdLDL-C and negatively correlates with HDL-C and adiponectin in patients with T2DM. In addition, when two groups were created on the basis of RemL-C, high RemL-C was found to correlate with FPG, HOMA-R, TC, TG, HDL-C, sdLDL-C, ACR, adiponectin, and ABI in patients with T2DM. Furthermore, sdLDL-C was found to be independently associated with high RemL-C in patients with T2DM. Because the clinical parameters associated with RemL-C are nearly identical to those associated with RLP-C and because RemL-C has been reported to be present at high concentration in patients with CAD [18], RemL-C, as well as RLP-C, may be a useful index in patients with T2DM who are at high risk of CVD.

A relationship between subclinical arteriosclerosis and RemL-C has also been previously reported. Taguchi et al. reported that RemL-C concentration positively correlates with max-IMT and mean-IMT in healthy women [33]. In the present study, RemL-C tended to positively correlate with max-IMT and mean-IMT (*χ^2^* = 3.140, *p* = 0.076 and *χ^2^* = 2.805, *p* = 0.094, respectively) in women with T2DM. Interestingly, multivariate logistic regression analysis indicated that ABI, which is a marker of PAD, is significantly and independently associated with high RemL-C in men with T2DM, suggesting that there may be a sex difference in the effect of remnant lipoproteins on the progression of arteriosclerosis. Further study is needed to clarify this.

Apo-AI, which is a major protein component of HDL-C, is thought to closely and inversely correlate with the risk of atherosclerotic disease [34]. Karagiannidis et al. reported that higher levels of Apo-AI are independent predictors of lower CAD complexity in T2DM [35]. In the present study, we have shown that a high Apo-AI concentration is an independent predictor of low RemL-C. These findings may indicate that RemL-C is linked to the complexity of CAD. In addition, we have shown that a high Apo-B concentration is an independent predictor of high RemL-C. Given that an association between Apo-B and CVD has previously been reported (35), a relationship between RemL-C and CVD may be revealed by future prospective studies.

In recent years, the relationship between TG-rich remnant lipoprotein and chronic kidney disease has been attracting attention. Sonoda et al. reported that the RemL-C concentration positively correlates with the ACR and inversely correlates with eGFR [36]. In contrast, we found that RemL-C did not significantly correlate with eGFR (*χ*^2^ = 0.20, *p* = 0.887). The reason for the discrepancy is unknown, but it may be explained by differences in patient background. RemL-C concentration was found to significantly positively correlate with ACR (*χ^2^* = 4.311, *p* = 0.038) in the present study, as well as the previous one. Relevant to this, Jun et al. demonstrated in a meta-analysis that fibrates reduce the risk of progression of albuminuria by 14% (*p* = 0.028) [37]. Moreover, Tsunoda et al. reported that fenofibrate reduces the concentration of RLP-C in patients with T2DM [29]. Thus, it is possible that RemL-C is associated with albuminuria and that a therapeutic intervention aimed at reducing RemL-C would help to prevent the progression of diabetic nephropathy in patients with T2DM.

Diabetes is associated not only with lipid metabolic disorders but also with higher concentrations of modified lipoproteins that are not routinely measured in clinical practice. Recently, the role of nitrated lipoproteins in the development of cardiovascular dysfunction in patients with diabetes has been discussed [38]. Nitrotyrosine (NT-Tyr) is a product of tyrosine modification by peroxynitrite, a potent prooxidant produced by the interaction of superoxide anions with nitric oxide [38]. The NT-Tyr concentration has been shown to be significantly higher in patients with T2DM than in non-diabetic individuals, and histopathological studies carried out to date on sections of arterial wall have shown that the NT-Tyr concentration is higher in patients with poor cardiovascular status [38]. Lipoproteins can undergo myeloperoxidase-catalyzed enzymatic nitration, and the reaction involves the apolipoproteins Apo-AI in HDL particles and Apo-B in LDL particles [38]. Because Apo-B is also present in remnant lipoprotein particles, the relationship between nitrated remnant lipoproteins and CVD risk is being clarified, and the establishment of a method for the measurement of the nitrated remnant lipoprotein concentration may be expected in the future.

The present study had several limitations. First, it was a single-center cross-sectional study with a relatively small number of participants. Thus, further large-scale prospective studies are needed to confirm the associations between RemL-C and clinical parameters in patients with T2DM. Second, because a direct comparison between RemL-C and RLP-C was not possible because of a lack of RLP-C measurements, it is difficult to discuss the factors affecting the relationships of RemL-C and RLP-C with CVD. However, Yoshida et al. reported that there is generally a significant correlation between RemL-C and RLP-C in individuals without diabetes or hypertension and not taking medications for hyperlipidemia, diabetes, or hypertension (r = 0.853, *p* < 0.0001) [26]. Moreover, they demonstrated that the RemL-C assay is likely to reflect the intermediate-density lipoprotein concentration more closely than RLP-C [26]. Therefore, the benefits of RemL-C measurement should be investigated further. Third, in the present study, other factors related to CVD, such as dyspnea classification and the heart failure status, were not available. It is a major limitation of the study. Further studies should address these issues in the future.

## 5. Conclusions

In the present study, we have demonstrated in T2DM that (i) RemL-C is associated with apolipoproteins related with remnant lipoproteins, (ii) high RemL-C concentration is associated with several factors related to lipid and glucose metabolism, (iii) sdLDL-C are independently associated with a high RemL-C concentration in both female and male subjects, (iv) ABI are independently associated with a high RemL-C concentration in male subjects, and (v) there was no difference in the RemL-C concentration between individuals with or without a history of CVD, possibly due to statin use. Therefore, RemL-C concentration may be a useful index of lipid and glucose metabolism, and that may be a marker of PAD in male subjects with T2DM. Regarding RemL-C as a risk factor for CVD, further prospective observational studies are needed in the future.

## Figures and Tables

**Figure 1 biomolecules-13-00468-f001:**
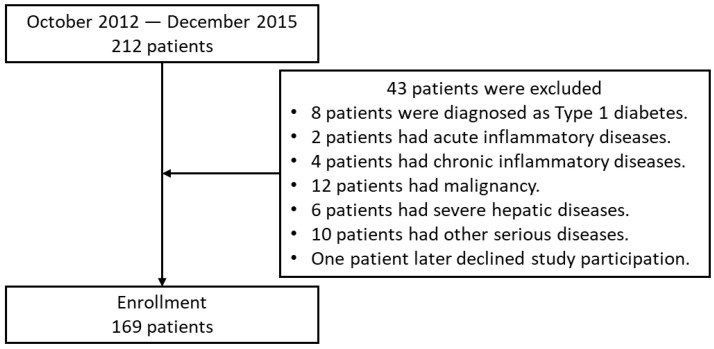
Flow chart of patient enrollment.

**Table 1 biomolecules-13-00468-t001:** Characteristics of the participants as a whole (*n* = 169) and of those with lower (Lo RemL-C (<0.24 mmol/L): *n* = 83) or higher (Hi RemL-C (≥0.24 mmol/L): *n* = 86) RemL-C concentrations.

	Unit	All	Lo RemL-C	Hi RemL-C	*p*
Age	years	61.2 ± 1.0	63.4 ± 1.2	59.0 ± 1.5	0.075
Sex	% female	42.0	45.8	38.3	0.296
Dur. of diabetes	years	10.3 ± 0.7	10.3 ± 1.0	10.3 ± 1.0	0.929
BMI	kg/m^2^	25.7 ± 0.3	25.7 ± 0.5	25.6 ± 0.5	0.471
SBP	mmHg	129.7 ± 1.4	130.4 ± 2.0	129.1 ± 2.1	0.547
DBP	mmHg	76.2 ± 1.0	74.7 ± 1.4	77.7 ± 1.3	0.151
FPG	mmol/L	7.29 ± 0.17	6.85 ± 0.22	7.72 ± 0.23	0.002
FPI	µU/mL	10.1 ± 0.5	9.0 ± 0.7	11.2 ± 0.8	0.071
HOMA-R		3.24 ± 0.20	2.69 ± 0.23	3.80 ± 0.33	0.010
HbA1c	%	7.51 ± 0.13	7.23 ± 0.16	7.78 ± 0.19	0.132
TC	mmol/L	4.83 ± 0.07	4.58 ± 0.09	5.07 ± 0.10	<0.001
TG	mmol/L	1.74 ± 0.08	1.12 ± 0.04	2.34 ± 0.13	<0.001
HDL-C	mmol/L	1.36 ± 0.03	1.51 ± 0.04	1.22 ± 0.04	<0.001
LDL-C	mmol/L	2.67 ± 0.06	2.56 ± 0.08	2.77 ± 0.10	0.098
sdLDL-C	mmol/L	0.81 ± 0.03	0.61 ± 0.03	1.00 ± 0.04	<0.001
RemL-C	mmol/L	0.29 ± 0.02	0.15 ± 0.01	0.42 ± 0.03	<0.001
Lp(a)	mg/dL	21.9 ± 2.0	23.3 ± 3.1	20.6 ± 2.5	0.902
hs-CRP	mg/dL	0.28 ± 0.06	0.35 ± 0.10	0.22 ± 0.05	0.243
eGFR	ml/min/1.73 m^2^	69.6 ± 1.5	70.1 ± 2.0	69.1 ± 2.1	0.887
ACR	mg/gCre	85.9 ± 19.1	67.7 ± 26.4	103.0 ± 27.7	0.038
Adiponectin	µg/mL	11.1 ± 0.6	13.4 ± 1.0	8.9 ± 0.7	<0.001
TNF-α	pg/mL	3.7 ± 0.7	3.1 ± 0.7	4.3 ± 1.2	0.312
IL-6	pg/mL	5.4 ± 0.7	5.9 ± 1.3	4.9 ± 0.7	0.333
baPWV	cm/s	1713.2 ± 27.1	1713.0 ± 36.5	1713.5 ± 40.3	0.816
ABI		1.14 ± 0.01	1.16 ± 0.02	1.12 ± 0.01	0.032
Mean CCA-IMT	mm	0.74 ± 0.02	0.75 ± 0.02	0.73 ± 0.02	0.254
Max CCA-IMT	mm	1.06 ± 0.03	1.10 ± 0.05	1.02 ± 0.04	0.108
Hypertension	%	59.8	62.7	60.0	0.491
Hyperlipidemia	%	66.9	60.2	80.0	0.090
CVD	%	13.0	15.7	10.5	0.300
Diabetes medication					
Oral agents	%	74.6	78.3	70.9	0.290
Sulfonylurea	%	34.5	31.7	37.2	0.453
Glinide	%	3.0	4.9	1.2	0.144
αGI	%	14.3	12.2	16.3	0.449
Biguanide	%	33.3	37.8	29.1	0.230
Thiazolidine	%	10.1	18.3	2.3	<0.001
DPP-4 inhibitor	%	47.6	50.0	45.4	0.546
SGLT2 inhibitor	%	0.6	0.0	1.2	0.246
GLP-1RA	%	3.6	3.6	3.5	0.953
Insulin	%	13.0	10.8	15.1	0.425
Statins	%	33.7	43.4	24.4	0.007
Fibrate	%	3.0	4.8	1.2	0.144
Eicosapentaenoic acid	%	3.0	4.8	1.2	0.144
ARBs or ACEIs	%	42.0	44.6	39.5	0.464
CCBs	%	37.3	41.0	33.7	0.381
Ant-platelet agents	%	19.5	20.5	18.6	0.729

Data are means ± SEM. BMI, body mass index; SBP, systolic blood pressure; DBP, diastolic blood pressure; FPG, fasting plasma glucose; FPI, fasting plasma insulin; TC, total cholesterol; TG, triglyceride; HDL-C, high-density lipoprotein cholesterol; LDL-C, low-density lipoprotein cholesterol; sdLDL-C, small dense LDL cholesterol; Rem-C, remnant cholesterol; hsCRP, high-sensitive C reactive protein; TNF-α, tumor necrosis factor-α; IL-6, interleukin-6; baPWV, brachial ankle pulse wave velocity; ABI, ankle brachial pressure index; CCA-IMT, intima-media thickness of common carotid artery; CVD, cardiovascular disease; αGI, α-glucosidase inhibitor; DPP-4, dipeptidyl peptidase-4; GLP-1RA, glucagon-like peptide-1 receptor agonists; ARBs, angiotensin II receptor blockers; ACEIs, angiotensin converting enzymes; CCBs, calcium channel blockers.

**Table 2 biomolecules-13-00468-t002:** Relationships between RemL-C and clinical parameters in patients with T2DM.

	Univariate	
	*r*	*p*
Age	−0.111	0.150
Dur. of DM	−0.050	0.521
BMI	−0.009	0.903
SBP	0.012	0.875
DBP	0.087	0.260
FPG	0.245	0.001
FPI	0.253	0.002
HOMA-R	0.322	<0.001
HbA1c	0.253	0.001
TC	0.394	<0.001
TG	0.867	<0.001
HDL-C	−0.395	<0.001
LDL-C	0.153	0.047
sdLDL-C	0.682	<0.001
Lp(a)	−0.024	0.762
hs-CRP	−0.008	0.924
eGFR	−0.009	0.903
ACR	0.121	0.127
Adiponectin	−0.302	<0.001
TNF-α	0.137	0.081
IL-6	0.023	0.770
baPWV	0.005	0.945
ABI	−0.060	0.443
meanCCA-IMT	−0.017	0.833
maxCCA-IMT	−0.043	0.584

Dur., duration; BMI, body mass index; SBP, systolic blood pressure; DBP, diastolic blood pressure; FPG, fasting plasma glucose; FPI, fasting plasma insulin; HOMA-R, homeostasis model assessment for insulin resistance; TC, total cholesterol; TG, triglyceride; HDL-C, high-density lipoprotein cholesterol; LDL-C, low-density lipoprotein cholesterol; sdLDL-C, small dense LDL cholesterol; RemL-C, remnant lipoprotein cholesterol; Lp(a), lipoprotein(a); hsCRP, high-sensitive C reactive protein; eGFR, estimated glomerular filtration rate; ACR, uremic albumin creatinine ratio; TNF-α, tumor necrosis factor-α; IL-6, interleukin-6; baPWV, brachial ankle pulse wave velocity; ABI, ankle brachial pressure index; CCA-IMT, intima-media thickness of common carotid artery.

**Table 3 biomolecules-13-00468-t003:** Univariate and multivariate stepwise regression analysis for apolipoproteins on RemL-C in patients with T2DM.

	Univariate		Multivariate		
	*r*	*p*	β	*p*	95% CI (Upper/Lower)
Apo-AI	−0.073	0.36	−0.232	<0.001	−0.004/−0.002
Apo-AII	0.123	0.12			
Apo-B	0.561	<0.001	0.289	<0.001	0.002/0.005
Apo-CII	0.609	<0.001			
Apo-CIII	0.623	<0.001	0.480	<0.001	0.027/0.046
Apo-E	0.547	<0.001	0.211	0.001	0.018/0.069

Apo, apolipoprotein; RemL-C, remnant lipoprotein cholesterol.

**Table 4 biomolecules-13-00468-t004:** Logistic regression analysis for the factors for lower or higher RemL-C levels in all subjects with T2DM.

	Univariate		Multivariate		
	*χ^2^*	*p*	Odds	*p*	95% CI (Upper/Lower)
HOMA-R	6.599	0.010			
sdLDL-C	42.460	<0.001	4363.290	<0.001	99,709.74/190.937
eGFR	0.020	0.887			
ACR	4.311	0.038			
Adiponectin	12.274	<0.001			
ABI	4.576	0.032	0.003	0.023	0.594/0.000

RemL-C, remnant lipoprotein cholesterol; HOMA-R, homeostasis model assessment for insulin resistance; sdLDL-C, small dense LDL cholesterol; eGFR, estimated glomerular filtration rate; ACR, uremic albumin creatinine ratio; ABI, ankle brachial pressure index. Factors included in the logistic regression analysis for lower or higher level of RemL-C were ABI, HOMA-R, sdLDL-C, adiponectin, ACR and eGFR. Estimates are for the following log odds: high RemL-C (≥0.24 mmol/L)/low RemL-C (<0.24 mmol/L).

**Table 5 biomolecules-13-00468-t005:** Logistic regression analyses for the factors of RemL-C in male participants with T2DM.

	Univariate		Multivariate		
	*χ^2^*	*p*	Odds	*p*	95% CI (Upper/Lower)
Age	0.384	0.535			
Dur. of DM	1.967	0.161			
BMI	0.028	0.868			
SBP	0.229	0.632			
DBP	0.630	0.427			
FPG	2.122	0.145			
FPI	3.782	0.052			
HOMA-R	3.843	0.050			
HbA1c	0.145	0.703			
TC	9.243	0.002			
TG	48.804	<0.001			
HDL-C	6.006	0.014			
LDL-C	2.218	0.136			
sdLDL-C	36.001	<0.001	11,627.31	<0.001	742,604.1/182.054
Lp(a)	0.054	0.817			
hs-CRP	0.468	0.494			
eGFR	0.089	0.765			
ACR	2.384	0.123			
Adiponectin	2.462	0.117			
TNF-α	0.058	0.810			
IL-6	1.321	0.250			
baPWV	0.061	0.805			
ABI	4.092	0.043	0.05	0.025	0.794/0.003
meanCCA-IMT	0.244	0.622			
maxCCA-IMT	0.769	0.380			
Hypertension	0.396	0.374			
Hyperlipidemia	3.069	0.080			
Diabetic microangiopathy					
Retinopathy	0.476	0.490			
Neuropathy	0.000	0.992			
Nephropathy	0.393	0.531			
CVD	0.096	0.756			

Dur., duration; BMI, body mass index; SBP, systolic blood pressure; DBP, diastolic blood pressure; FPG, fasting plasma glucose; FPI, fasting plasma insulin; HOMA-R, homeostasis model assessment for insulin resistance; TC, total cholesterol; TG, triglyceride; HDL-C, high-density lipoprotein cholesterol; LDL-C, low-density lipoprotein cholesterol; sdLDL-C, small dense LDL cholesterol; RemL-C, remnant lipoprotein cholesterol; Lp(a), lipoprotein(a); hsCRP, high-sensitive C reactive protein; eGFR, estimated glomerular filtration rate; ACR, uremic albumin creatinine ratio; TNF-α, tumor necrosis factor-α; IL-6, interleukin-6; baPWV, brachial ankle pulse wave velocity; ABI, ankle brachial pressure index; CCA-IMT, intima-media thickness of common carotid artery; CVD, cardiovascular disease. Logistic regression analyses included HOMA-R, age, sdLDL-C, adiponectin and ABI. Estimates are for the following log odds: high RemL-C (≥0.24 mmol/L)/low RemL-C (<0.24 mmol/L).

**Table 6 biomolecules-13-00468-t006:** Logistic regression analyses for the factors of RemL-C in female participants with T2DM.

	Univariate		Multivariate		
	*χ^2^*	*p*	Odds	*p*	95% CI (Upper/Lower)
Age	4.434	0.035			
Dur. of DM	1.990	0.158			
BMI	0.758	0.384			
SBP	0.294	0.588			
DBP	1.440	0.230			
FPG	7.219	0.007			
FPI	0.811	0.368			
HOMA-R	3.401	0.065			
HbA1c	3.826	0.051			
TC	3.341	0.068			
TG	38.050	<0.001			
HDL-C	22.799	<0.001			
LDL-C	0.689	0.407			
sdLDL-C	15.098	<0.001	708.125	<0.001	64,330.18/7.795
Lp(a)	0.012	0.913			
hs-CRP	0.815	0.367			
eGFR	0.190	0.663			
ACR	1.413	0.235			
Adiponectin	10.873	<0.001			
TNF-α	1.006	0.316			
IL-6	0.098	0.754			
baPWV	0.436	0.509			
ABI	1.350	0.245			
meanCCA-IMT	2.805	0.094			
maxCCA-IMT	3.140	0.076			
Hypertension	0.000	0.995			
Hyperlipidemia	0.473	0.492			
Diabetic microangiopathy					
Retinopathy	6.652	0.010			
Neuropathy	3.940	0.047			
Nephropathy	2.237	0.135			
CVD	1.024	0.312			

Dur., duration; BMI, body mass index; SBP, systolic blood pressure; DBP, diastolic blood pressure; FPG, fasting plasma glucose; FPI, fasting plasma insulin; HOMA-R, homeostasis model assessment for insulin resistance; TC, total cholesterol; TG, triglyceride; HDL-C, high-density lipoprotein cholesterol; LDL-C, low-density lipoprotein cholesterol; sdLDL-C, small dense LDL cholesterol; RemL-C, remnant lipoprotein cholesterol; Lp(a), lipoprotein(a); hsCRP, high-sensitive C reactive protein; eGFR, estimated glomerular filtration rate; ACR, uremic albumin creatinine ratio; TNF-α, tumor necrosis factor-α; IL-6, interleukin-6; baPWV, brachial ankle pulse wave velocity; ABI, ankle brachial pressure index; CCA-IMT, intima-media thickness of common carotid artery; CVD, cardiovascular disease. Logistic regression analyses included ABI, HOMA-R, sdLDL-C, adiponectin, and age. Estimates are for the following log odds: high RemL-C (≥0.24 mmol/L)/low RemL-C (<0.24 mmol/L).

**Table 7 biomolecules-13-00468-t007:** Univariate analyses of clinical parameters between the participants with and without a history of CVD in patients with T2DM.

	Unit	CAD (−)	CAD (+)	*F*	*p*
Age	years	59.96	68.22	8.510	0.004
Dur. of DM	years	9.26	17.09	15.402	<0.001
BMI	kg/m^2^	25,48	27.02	2.313	0.130
SBP	mmHg	129.35	132.17	0.448	0.504
DBP	mmHg	76.71	72.96	1.776	0.184
FPG	mmol/L	130.79	136.22	0.389	0.534
FPI	µU/mL	9.65	13.57	5.697	0.018
HOMA-R		3.14	4.10	2.178	0.142
HbA1c	%	7.53	7.45	0.054	0.817
TC	mmol/L	4.94	4.17	14.932	<0.001
TG	mmol/L	1.74	1.77	0.014	0.907
HDL-C	mmol/L	1.38	1.29	1.026	0.313
LDL-C	mmol/L	2.77	2.08	15.314	<0.001
sdLDL-C	mmol/L	0.83	0.70	2.498	0.116
RemL-C	mmol/L	0.29	0.25	0.732	0.394
Lp(a)	mg/dL	18.94	38.80	12.71	<0.001
hs-CRP	mg/dL	0.315	0.103	1.549	0.204
eGFR	ml/min/1.73 m^2^	71.61	56.43	13.333	<0.001
ACR	mg/gCre	83.48	103.77	0.130	0.719
Adiponectin	µg/mL	11.43	9.18	1.434	0.233
TNF-α	pg/mL	3.88	2.99	0.192	0.662
IL-6	pg/mL	5.64	4.04	0.543	0.462
baPWV	cm/s	1709.2	1729.1	0.063	0.803
ABI		1.143	1.092	4.264	0.041
meanCCA-IMT	mm	0.717	0.858	10.867	0.001
maxCCA-IMT	mm	1.018	1.332	12.005	<0.001

Dur., duration; BMI, body mass index; SBP, systolic blood pressure; DBP, diastolic blood pressure; FPG, fasting plasma glucose; FPI, fasting plasma insulin; HOMA-R, homeostasis model assessment for insulin resistance; TC, total cholesterol; TG, triglyceride; HDL-C, high-density lipoprotein cholesterol; LDL-C, low-density lipoprotein cholesterol; sdLDL-C, small dense LDL cholesterol; RemL-C, remnant lipoprotein cholesterol; Lp(a), lipoprotein(a); hsCRP, high-sensitive C reactive protein; eGFR, estimated glomerular filtration rate; ACR, uremic albumin creatinine ratio; TNF-α, tumor necrosis factor-α; IL-6, interleukin-6; baPWV, brachial ankle pulse wave velocity; ABI, ankle brachial pressure index; meanCCA-IMT, mean intima-media thickness of common carotid artery; maxCCA-IMT, maximum intima-media thickness of common carotid artery; CVD, cardiovascular disease.

## Data Availability

The datasets generated during and/or analyzed during the current study are available from the corresponding author on reasonable request.

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
