# Peer review of "Significance of Circulating Remnant Lipoprotein Cholesterol Levels Measured by Homogeneous Assay in Patients with Type 2 Diabetes"

_biomolecules, 2023, doi:10.3390/biom13030468_

Round 1
Reviewer 1 Report
The authors aimed to characterize the relationships between RemL-C and several parameters related to CVD in patients with diabetes, and to assess the utility of RemL-C measurement for patients with T2DM. The manuscript is well presented. However, I have the following comments:
1- The authors performed a cross-sectional study of 212 patients with T2DM who were hospitalized to improve their glycemic control at Kumamoto University Hospital between 2012 and 2014. Data from a total of 169 patients with T2DM (98 men and 71 women) were analyzed. I did not see any flow-chart of this study in the current manuscript. Furthermore, the authors explained that a remnant lipoprotein cholesterol homogeneous assay (RemL-C assay; Kyowa Medex, Japan) has been recently developed that uses a special surfactant [polyoxyethylene-polyoxybutylene (POE-POB) block copolymer] and phospholipase D, which selectively solubilizes and degrades TG-rich remnant lipoproteins, very low-density lipoprotein (VLDL) remnants, and chylomicron remnants. However, how was this new method applied to a population which was screened between 2012 and 2014?
2- In contrast to the RLP-C assay, the RemL-C assay can be performed on universal automated analyzers, which enables rapid, high-throughput measurements. Is it the unique advantage of this technology? If yes, I do not consider that it is a major progress.
3- In the discussion section, the authors explained that this is the first study to investigate the relationships between RemL-C and clinical parameters related to CVD risk in patients with T2DM. I did not find any clinical parameter in this study, like dyspnea classification, heart failure, or other clinical complications of T2DM.
Author Response
#1-1 Flow-chart of this study
I did not see any flow-chart of this study in the current manuscript.
Thank you for your suggestion. According to your suggestion, we added the flowchart of this study in “Figure 1” of the revised manuscript. Regarding this, we found some mistakes in the periods of recruitment. Therefore, we rewrote some parts of the contents in the “Materials and Methods” section of the revised manuscript (Page 2, lines 90-92).
Revised manuscript (Page 2, lines 90-92)
- Materials and Methods
2.1 Participants
“We performed a cross-sectional study of 212 patients with T2DM who were hospitalized to improve their glycemic control at Kumamoto University Hospital between October 2012 and December 2015.”
#1-2 The way to apply the RemL-C assay for the subjects
How was this new method (RemL-C assay) applied to a population which was screened between 2012 and 2014?
Thank you for your comment on the RemL-C assay. The RemL-C assay kit was purchased from Minaris Medical Co. ltd. and used for the subjects. We rewrote some parts of the contents in the “Materials and Methods” section of the revised manuscript (Page 3, lines 120-122).
Revised manuscript (Page 3, lines 120-122).
- Materials and Methods
2.3 Laboratory measurements
“Homogeneous methods were used to measure the RemL-C, high-density lipoprotein cholesterol (HDL-C) (Minaris Medical Co. Ltd., Tokyo, Japan), and small dense LDL-C (sdLDL) (Denka Seiken Co. ltd., Tokyo, Japan) concentrations.”
#2 The unique advantage of the RemL-C
In contrast to the RLP-C assay, the RemL-C assay can be performed on universal automated analyzers, which enables rapid, high-throughput measurements. Is it the unique advantage of this technology? If yes, I do not consider that it is a major progress.
Thank you for your suggestion. This method can be measured with an automated analyzer, does not require sample pretreatment, greatly shortens the measurement time (10 minutes), allows simultaneous measurement of multiple samples, reduces measurement errors, and improves measurement accuracy. In that sense, the RemL-C assay would be a more convenient method than the RLP-C assay. However, since it depends on the reader's point of view, we have not mentioned that in this article. To deepen the reader’s understanding of RemL-C assay, we added some characteristics of RemL-C assay in the “Discussion” section of the revised manuscript (Page 12, lines 324-327).
Revised manuscript (Page 12, lines 324-327)
- Discussion
“In contrast, the second method, the RemL-C assay, can be performed on a universal autoanalyzer, does not require sample pre-treatment, greatly shortens the measurement time (to 10 minutes), permits the simultaneous measurement of multiple samples, reduces measurement errors, and improves measurement accuracy.”
#3 Lack of several clinical parameter for CVD
In the discussion section, the authors explained that this is the first study to investigate the relationships between RemL-C and clinical parameters related to CVD risk in patients with T2DM. I did not find any clinical parameter in this study, like dyspnea classification, heart failure, or other clinical complications of T2DM.
Thank you for your suggestion. Unfortunately, as pointed out by the reviewers, other factors related to CVD such as dyspnea classification and heart failure are missing. Regarding this point, I added it to the limitation in the revised manuscript (Page 14, lines 433-435).
Revised manuscript (Page 14, lines 433-435)
- Discussion
“Third, in the present study, other factors related to CVD, such as dyspnea classification and the heart failure status, were not available. Further studies should address these issues in the future.”
Reviewer 2 Report
Dear authors,
your work is really interesting! Some minor comments might further improve your manuscript:
1. please elaborate on every abbreviation used upon their first use,
2. please provide p values in Table 1 for the comparison between high and low RemL-C groups,
3. Every finding of your analysis is in concordance with the existing literature. This is important! However, i think that you could discuss a little more the association of apolipoproteins with RemL-C since apolipoprotein A1 has been consistently associated with ameliorated lipidemic profile and decreased coronary artery disease complexity while apolipoprotein B has been linked with increased thrombus burden (https://cardiab.biomedcentral.com/articles/10.1186/s12933-022-01494-9)
4. Please provide further data on the diabetic therapy of your study participants, if available and make comparisons of the RemL-C levels and the medication groups.
Author Response
Reply to the Reviewer 2’s comments
#1 Abbreviations
Please elaborate on every abbreviation used upon their first use.
Thank you for your suggestion. According to your suggestion, we checked all abbreviations and rewrote the wrong ones as follows. Especially, because of the word limitation, some parts of the contents were rewritten in the “Abstract” of the revised manuscript.
Revised manuscript
Abstract
“Remnant lipoproteins (RLs), which are typically present at high concentrations in patients with type 2 diabetes mellitus (T2DM), are associated with cardiovascular disease (CVD). Although an RL cholesterol homogeneous assay (RemL-C) is available for the measurement of RL concentrations, there have been no studies of the relationship between RemL-C and clinical parameters in T2DM. Therefore, we evaluated the relationships between RemL-C and CVD-related parameters in patients with T2DM. We performed a cross-sectional study of 169 patients with T2DM who were hospitalized at Kumamoto University Hospital. Compared with those with low RemL-C, those with higher RemL-C had higher fasting plasma glucose, homeostasis model assessment for insulin resistance (HOMA-R), total cholesterol, triglyceride, small dense low-density lipoprotein cholesterol (sdLDL-C), and urinary albumin-creatinine ratio; and lower high-density lipoprotein cholesterol, adiponectin, and ankle brachial pressure index (ABI). Multivariate logistic regression analysis showed that sdLDL-C and ABI were significantly and independently associated with high RemL-C. Although LDL-C was lower in participants with CVD, there was no difference in RemL-C between participants with or without CVD. Thus, RemL-C may represent a useful index of RL particles in patients with T2DM, and further therapeutic interventions may be needed in the future, given that RemL-C represents a residual risk.”
Text
- type 2 diabetes mellitus (T2DM) (Page 2, line 48)
- glycated hemoglobin (HbA1c) (Page 2, line 52)
- remnant-like particle (RLP)-cholesterol (RLP-C) (Page 2, lines 65-66)
- apolipoprotein (Apo) (Page 2, lines 68-69)
- glutamic acid decarboxylase (GAD) (Page 2, line 94)
- systolic blood pressure (SBP) and diastolic blood pressure (DBP) (Page 3, line 103)
- “CKD” was rewritten to “chronic kidney disease” (Page 13, lines 394-395)
#2 P values of Table 1
Please provide p values in Table 1 for the comparison between high and low RemL-C groups.
Thank you for your suggestion. According to your suggestion, we added p values in Table 1.
#3 Association between apolipoprotein and RemL-C
I think that you could discuss a little more the association of apolipoproteins with RemL-C since apolipoprotein A1 has been consistently associated with ameliorated lipidemic profile and decreased coronary artery disease complexity while apolipoprotein B has been linked with increased thrombus burden (https://cardiab.biomedcentral.com/articles/10.1186/s12933-022-01494-9)
Thank you for your suggestion. According to your suggestion, we added several sentences in the “Discussion” section of the revised manuscript (Page 13, lines 385-393).
Revised manuscript (Page 13, lines 385-393)
- Discussion
“ApoAI, which is a major protein component of HDL-C, is thought to closely and inversely correlate with the risk of atherosclerotic disease [34]. Karagiannidis et al. reported that higher levels of Apo-AI are independent predictors of lower CAD complexity in T2DM [35]. In the present study, we have shown that a high Apo-AI concentration is an independent predictor of low RemL-C. These findings may indicate that RemL-C is linked to the complexity of CAD. In addition, we have shown that a high Apo-B concentration is an independent predictor of high RemL-C. Given that an association between Apo-B and CVD has previously been reported (35), a relationship be-tween RemL-C and CVD may be revealed by future prospective studies.”
#4 Diabetic therapy and RemL-C
Please provide further data on the diabetic therapy of your study participants, if available and make comparisons of the RemL-C levels and the medication groups.
Thank you for your request. Regarding drug therapy, higher RemL-C had a significantly lower number of participants treated with thiazolidine or statins. However, including thiazolidine use and statin use in the model for multivariate logistic regression analysis had no effect on the results of the analysis.
We added several data on diabetic therapy in “Table 1”, and added some sentences in the “Results” section of the revised manuscript (Page 4, lines 181-183., Page 7, lines 235-236).
Revised manuscript (Page 4, lines 181-183., Page 7, lines 235-236)
- Results
3.1. Characteristics of the participants
“Regarding drug therapy, high RemL-C concentrations were present in significantly fewer participants being treated with a thiazolidinedione or a statin (Table 1).”
“The inclusion of thiazolidine use and statin use in the multivariate logistic regression analysis had no effect on the results (data not shown).”
Reviewer 3 Report
I received an original research article for review entitled " Relationship between remnant circulating lipoprotein cholesterol concentration, measured by homogeneous assay, and clinical parameters in patients with type 2 diabetes", prepared by Kazumi Matsushima-Nagata, which was submitted to the Biomolecules (IF=6.064). The article concerns a very important issue, which is some aspects of dyslipidemia in patients with type 2 diabetes mellitus. Both dyslipidemia and type 2 diabetes mellitus belong to components of the metabolic syndrome, which is one of the most important problems for public health worldwide leading to a deterioration in the length and quality of life, therefore research in this area is of fundamental importance. The article is generally well prepared and represent some scientific value, but in my opinion significant modifications are necessary that may contribute to further improvement of the quality and attractiveness of the presented manuscript.
1) The authors use both the term coronary artery disease (CAD) and coronary heart disease (CHD). In my opinion, the two terms are synonymous and can be used interchangeably. If, in the Authors' opinion, these terms differ, please explain the difference. If these terms are synonymous, please choose one of these terms and use it consistently.
2) In the introduction, the authors, on the one hand, write about diseases of the cardiovascular system, but enumeratively indicate only ischemic heart disease. I believe that it is worth adding an additional paragraph in the introduction devoted to general information on cardiovascular disease in diabetic patients. It is worth pointing out that in the course of atherosclerosis, not only ischemic heart disease develops, but also cerebrovascular disease and peripheral arterial disease. Atherosclerosis in diabetic patients is slightly different, e.g. in the case of atherosclerosis of the arteries of the lower limbs, the changes are usually multilevel, the arteries below the knee are usually more involved, and diabetic patients are characterized by the presence of calcifications of the middle layer of the arterial wall. In atherosclerotic cardiovascular disease, invasive revascularization therapy, including percutaneous balloon angioplasty with optional stent implantation, is of great importance, but the phenomenon of restenosis significantly reduces the long-term effectiveness of such treatment and may contribute to the need for reinvention, and diabetes is associated with an increased risk of restenosis in the stent. (10.3390/ijerph182211970; 10.3390/ijerph191811242)
3) It is worth mentioning in the introduction or in the discussion that dyslipidemia is not only abnormal levels of lipid metabolism parameters, but also the presence of modified, dysfunctional lipoproteins. Recently, the role of nitrated lipoproteins in the development of cardiovascular dysfunction in diabetic patients has been discussed.
4) All abbreviations should be explained on first use, both in the abstract, in the main text, and in table and figure captions. The abbreviation T2DM is not explained in the main text of the paper. This should be corrected even though the meaning of this abbreviation is obvious.
5) The abbreviation HbA1c should be explained on first use (glycated hemoglobin).
6) The description of the statistical analysis should describe how the correspondence between the distributions of empirical quantitative variables and the normal distribution was examined. Were the empirical distributions of all quantitative variables normal?
7) All laboratory tests that are described in the results should also be described in the methodology section. The methodology of laboratory tests does not describe adiponectin, tumor necrosis factor alpha and interleukin 6. The study methodology also does not describe the determination of apolipoprotein levels. In conclusion, all laboratory tests and other additional tests that are described in the results should also be described in the methodology.
8) The list of references must be prepared in accordance with the requirements of the MDPI, even in the editorial and stylistic part.
9) English should be revised and verified by a philologist.
Author Response
Reply to the Reviewer 3’s comments
#1 difference between CHD and CAD
The authors use both the term coronary artery disease (CAD) and coronary heart disease (CHD). In my opinion, the two terms are synonymous and can be used interchangeably. If, in the Authors' opinion, these terms differ, please explain the difference. If these terms are synonymous, please choose one of these terms and use it consistently.
Thank you for your advice. We also think that these terms are synonymous. So, we unified to CAD. We rewrote CHD to CAD in the “Introduction” of the revised manuscript (Page 2, lines 45-54).
Revised manuscript (Page 2, line 45-54)
- Introduction
“Indeed, the Framingham Heart Study revealed that diabetes is an independent risk factor for CAD: its presence results in a 1.5-fold higher multivariate-adjusted risk of CAD in men and a 1.8-fold higher adjusted risk in women [7]. The Hisayama Study also demonstrated that type 2 diabetes mellitus (T2DM) significantly increases the risks of both cerebral infarction and CAD during a 5-year follow-up period in a general Japanese population [8]. In addition, the serum TG concentration is a key predictor of CAD, having comparable value to low-density lipoprotein (LDL)-cholesterol (LDL-C) and glycated hemoglobin (HbA1c) in Japanese patients with T2DM [9]. This suggests that remnant lipoproteins, which contain a large amount of TG, may also be a predictor of CAD in T2DM.”
#2
In the introduction, the authors, on the one hand, write about diseases of the cardiovascular system, but enumeratively indicate only ischemic heart disease. I believe that it is worth adding an additional paragraph in the introduction devoted to general information on cardiovascular disease in diabetic patients. It is worth pointing out that in the course of atherosclerosis, not only ischemic heart disease develops, but also cerebrovascular disease and peripheral arterial disease. Atherosclerosis in diabetic patients is slightly different, e.g. in the case of atherosclerosis of the arteries of the lower limbs, the changes are usually multilevel, the arteries below the knee are usually more involved, and diabetic patients are characterized by the presence of calcifications of the middle layer of the arterial wall. In atherosclerotic cardiovascular disease, invasive revascularization therapy, including percutaneous balloon angioplasty with optional stent implantation, is of great importance, but the phenomenon of restenosis significantly reduces the long-term effectiveness of such treatment and may contribute to the need for reinvention, and diabetes is associated with an increased risk of restenosis in the stent. (10.3390/ijerph182211970; 10.3390/ijerph191811242)
Thank you for your suggestion. We agree with your comment. We added several sentences and two references in the “Introduction” and “References” section of the revised manuscript (Page 2, lines 55-63., Reference #10, 11).
Revised manuscript (Page 2, lines 55-63., Reference #10, 11)
- Introduction
“During the course of atherosclerosis, cerebrovascular disease and peripheral arterial disease are observed, as well as the development of ischemic heart disease. [10]. In the case of atherosclerosis of the arteries of the lower extremities, this typically develops in multiple stages, the arteries distal to the knee are more commonly affected, and it is characterized by the presence of calcification of the middle layers of the arterial walls in patients with diabetes. Invasive revascularization therapy, including percutaneous balloon angioplasty with optional stent placement, is important for patients with CAD, but the phenomenon of restenosis, which is especially common in patients with diabetes, limits the long-term effectiveness of this treatment [11].”
#3
It is worth mentioning in the introduction or in the discussion that dyslipidemia is not only abnormal levels of lipid metabolism parameters, but also the presence of modified, dysfunctional lipoproteins. Recently, the role of nitrated lipoproteins in the development of cardiovascular dysfunction in diabetic patients has been discussed.
Thank you for your suggestion. According to your suggestion, we added several sentences in the “Discussion” section (from Page 13, lines 407-421) and added reference #38 in the “References” section of the revised manuscript.
Revised manuscript (from Page 13, lines 407-421)
- Discussion
“Diabetes is associated not only with lipid metabolic disorders but also with higher concentrations of modified lipoproteins that are not routinely measured in clinical practice. Recently, the role of nitrated lipoproteins in the development of cardiovascular dysfunction in patients with diabetes has been discussed [38]. Nitrotyrosine (NT-Tyr) is a product of tyrosine modification by peroxynitrite, a potent prooxidant produced by the interaction of superoxide anions with nitric oxide [38]. The NT-Tyr concentration has been shown to be significantly higher in patients with T2DM than in non-diabetic individuals, and histopathological studies carried out to date on sections of arterial wall have shown that the NT-Tyr concentration is higher in patients with poor cardiovascular status [38]. Lipoproteins can undergo myeloperoxidase-catalyzed enzymatic nitration, and the reaction involves the apolipoproteins Apo-AI in HDL particles and Apo-B in LDL particles [38]. Because Apo-B is also present in remnant lipoprotein particles, the relationship between nitrated remnant lipoproteins and CVD risk is being clarified, and the establishment of a method for the measurement of the nitrated remnant lipoprotein concentration may be expected in the future.”
#4 Abbreviations
All abbreviations should be explained on first use, both in the abstract, in the main text, and in table and figure captions. The abbreviation T2DM is not explained in the main text of the paper. This should be corrected even though the meaning of this abbreviation is obvious.
Thank you for your suggestion. According to your suggestion, we checked all abbreviations and rewrote the wrong ones as follows. Especially, because of the word limitation, some parts of the contents were rewritten in the “Abstract” of the revised manuscript.
Revised manuscript
Abstract
“Remnant lipoproteins (RLs), which are typically present at high concentrations in patients with type 2 diabetes mellitus (T2DM), are associated with cardiovascular disease (CVD). Although an RL cholesterol homogeneous assay (RemL-C) is available for the measurement of RL concentrations, there have been no studies of the relationship between RemL-C and clinical parameters in T2DM. Therefore, we evaluated the relationships between RemL-C and CVD-related parameters in patients with T2DM. We performed a cross-sectional study of 169 patients with T2DM who were hospitalized at Kumamoto University Hospital. Compared with those with low RemL-C, those with higher RemL-C had higher fasting plasma glucose, homeostasis model assessment for insulin resistance (HOMA-R), total cholesterol, triglyceride, small dense low-density lipoprotein cholesterol (sdLDL-C), and urinary albumin-creatinine ratio; and lower high-density lipoprotein cholesterol, adiponectin, and ankle brachial pressure index (ABI). Multivariate logistic regression analysis showed that sdLDL-C and ABI were significantly and independently associated with high RemL-C. Although LDL-C was lower in participants with CVD, there was no difference in RemL-C between participants with or without CVD. Thus, RemL-C may represent a useful index of RL particles in patients with T2DM, and further therapeutic interventions may be needed in the future, given that RemL-C represents a residual risk.”
Text
- type 2 diabetes mellitus (T2DM) (Page 2, line 48)
- glycated hemoglobin (HbA1c) (Page 2, line 52)
- remnant-like particle (RLP)-cholesterol (RLP-C) (Page 2, lines 65-66)
- apolipoprotein (Apo) (Page 2, lines 68-69)
- glutamic acid decarboxylase (GAD) (Page 2, line 94)
- systolic blood pressure (SBP) and diastolic blood pressure (DBP) (Page 3, line 103)
- “CKD” was rewritten to “chronic kidney disease” (Page 13, lines 394-395)
#5 The abbreviation of HbA1c
The abbreviation HbA1c should be explained on first use (glycated hemoglobin).
Thank you for your suggestion. According to your suggestion, we added the full spell of HbA1c in the “Introduction” of the revised manuscript (Page 2, line 52).
Revised manuscript (Page 2, line 52)
- Introduction
“glycated hemoglobin (HbA1c)”
#6 The methodology of laboratory tests
The description of the statistical analysis should describe how the correspondence between the distributions of empirical quantitative variables and the normal distribution was examined. Were the empirical distributions of all quantitative variables normal?
Thank you for your suggestion. Regarding continuous variables, some variables showed skewed distribution (i.e., duration of diabetes, HOMA-R, TG, hs-CRP, RemL-C, Lp(a), adiponectin, TNF-α, IL-6, eGFR, and ACR), so they were logarithmically transformed before the analysis. We added several sentences in the “Materials and Methods” section of the revised manuscript (Page 4, lines 171-174).
Revised manuscript (Page 4, lines 171-174)
- Materials and Methods
2.5. Statistical analyses
“Because some of the variables showed skewed distributions (duration of diabetes, HOMA-R, TG, hs-CRP, RemL-C, Lp(a), adiponectin, TNF-α, IL-6, eGFR, and ACR), these data were logarithmically transformed before this analysis.”
#7 The methodology of laboratory tests
All laboratory tests that are described in the results should also be described in the methodology section. The methodology of laboratory tests does not describe adiponectin, tumor necrosis factor alpha and interleukin 6. The study methodology also does not describe the determination of apolipoprotein levels. In conclusion, all laboratory tests and other additional tests that are described in the results should also be described in the methodology.
We apologize to be lack of several methods of laboratory tests in the original manuscript. According to your suggestion, we rewrote the study methodology in the “Materials and Methods” section of the revised manuscript (from Page 3, line 119 to Page 4, line 137).
Revised manuscript (from Page 3, line 119 to Page 4, line 135)
- Materials and Methods
2.3. Laboratory measurements
“Enzymatic methods (Minaris Medical Co. ltd., Tokyo, Japan) were used to measure the serum TC and TG concentrations. Homogeneous methods were used to measure the RemL-C, high-density lipoprotein cholesterol (HDL-C) (Minaris Medical Co. Ltd., Tokyo, Japan), and small dense LDL-C (sdLDL) (Denka Seiken Co. ltd., Tokyo, Japan) concentrations. Immunoturbidimetric methods (Sekisui Medical Co. ltd., Tokyo, Japan) were used to measure the Apo-AI, Apo-AII, Apo-B, Apo-CIII, and Apo-E concentrations. The enzymatic and immunoturbidimetric measurements were performed on a Hitachi 7180 Auto Analyzer (Hitachi High-Tech Corporation, Tokyo, Japan). Latex aggregation methods (Denka Seiken Co. ltd., Tokyo, Japan) were used for the measurement of lipoprotein(a) (Lp(a)) and adiponectin concentrations on the Hitachi 7180 Auto Analyzer. An oxygen electrode method (Arkray, Inc., Kyoto, Japan) was used to measure the fasting plasma glucose (FPG) concentration on a GA1170. An ion exchange chromatography method (Tosoh Corporation, Tokyo, Japan) was used to quantify HbA1c on an HLC-732 G8. An electrochemiluminescence method (Roche Diagnostics K.K., Tokyo, Japan) was used to measure fasting plasma insulin (FPI) concentration on a Modular Analytics E module. A latex aggregation method (Denka Seiken Co. ltd., Tokyo, Japan) was used to measure high-sensitivity C-reactive protein (hsCRP) concentration on a BM2250. The measurement of the interleukin-6 (IL-6) and tumor necrosis factor-α (TNF-α) concentrations was outsourced to SRL, Inc (Tokyo, Japan).”
#8 References
The list of references must be prepared in accordance with the requirements of the MDPI, even in the editorial and stylistic part.
Thank you for your advice. According to the requirements of the MDPI, we have corrected all references.
#9 Revised English by a philologist
English should be revised and verified by a philologist.
Thank you for your suggestion. The first version of the manuscript was already edited by an English proofreading company. However, the revised manuscript was edited again by the English proofreading company. The fact is written in the “Acknowledgments” of the revised manuscript (Page 14, lines 462-463). In addition, an English proofreading certificate is attached.

Round 2
Reviewer 1 Report
#1-1 Flow-chart of this study
I did not see any flow-chart of this study in the current manuscript.
Thank you for your suggestion. According to your suggestion, we added the flowchart of this study in “Figure 1” of the revised manuscript. Regarding this, we found some mistakes in the periods of recruitment. Therefore, we rewrote some parts of the contents in the “Materials and Methods” section of the revised manuscript (Page 2, lines 90-92).
>It is one year! It is a large mistake.
Revised manuscript (Page 2, lines 90-92)
- Materials and Methods
2.1 Participants
“We performed a cross-sectional study of 212 patients with T2DM who were hospitalized to improve their glycemic control at Kumamoto University Hospital between October 2012 and December 2015.”
#1-2 The way to apply the RemL-C assay for the subjects
How was this new method (RemL-C assay) applied to a population which was screened between 2012 and 2014?
Thank you for your comment on the RemL-C assay. The RemL-C assay kit was purchased from Minaris Medical Co. ltd. and used for the subjects. We rewrote some parts of the contents in the “Materials and Methods” section of the revised manuscript (Page 3, lines 120-122).
>That means that the The RemL-C assay kit already exist in 2012? Is it really new?
Revised manuscript (Page 3, lines 120-122).
- Materials and Methods
2.3 Laboratory measurements
“Homogeneous methods were used to measure the RemL-C, high-density lipoprotein cholesterol (HDL-C) (Minaris Medical Co. Ltd., Tokyo, Japan), and small dense LDL-C (sdLDL) (Denka Seiken Co. ltd., Tokyo, Japan) concentrations.”
#2 The unique advantage of the RemL-C
In contrast to the RLP-C assay, the RemL-C assay can be performed on universal automated analyzers, which enables rapid, high-throughput measurements. Is it the unique advantage of this technology? If yes, I do not consider that it is a major progress.
Thank you for your suggestion. This method can be measured with an automated analyzer, does not require sample pretreatment, greatly shortens the measurement time (10 minutes), allows simultaneous measurement of multiple samples, reduces measurement errors, and improves measurement accuracy. In that sense, the RemL-C assay would be a more convenient method than the RLP-C assay. However, since it depends on the reader's point of view, we have not mentioned that in this article. To deepen the reader’s understanding of RemL-C assay, we added some characteristics of RemL-C assay in the “Discussion” section of the revised manuscript (Page 12, lines 324-327).
>I still not have the same point of view.
Revised manuscript (Page 12, lines 324-327)
- Discussion
“In contrast, the second method, the RemL-C assay, can be performed on a universal autoanalyzer, does not require sample pre-treatment, greatly shortens the measurement time (to 10 minutes), permits the simultaneous measurement of multiple samples, reduces measurement errors, and improves measurement accuracy.”
#3 Lack of several clinical parameter for CVD
In the discussion section, the authors explained that this is the first study to investigate the relationships between RemL-C and clinical parameters related to CVD risk in patients with T2DM. I did not find any clinical parameter in this study, like dyspnea classification, heart failure, or other clinical complications of T2DM.
Thank you for your suggestion. Unfortunately, as pointed out by the reviewers, other factors related to CVD such as dyspnea classification and heart failure are missing. Regarding this point, I added it to the limitation in the revised manuscript (Page 14, lines 433-435).
Revised manuscript (Page 14, lines 433-435)
- Discussion
“Third, in the present study, other factors related to CVD, such as dyspnea classification and the heart failure status, were not available. Further studies should address these issues in the future.”
> I think that it is a major limitation of this study.
Author Response
Response to Reviewer 1 Comments
Point 1: Original comment from reviewer 1 is that “I did not see any flow-chart of this study in the current manuscript.”
Reply from authors is that “Thank you for your suggestion. According to your suggestion, we added the flowchart of this study in “Figure 1” of the revised manuscript. Regarding this, we found some mistakes in the periods of recruitment. Therefore, we rewrote some parts of the contents in the “Materials and Methods” section of the revised manuscript (Page 2, lines 90-92).”
Second comment from reviewer 1 is that “It is one year! It is a large mistake.”
Response 1: I’m so sorry to make such large mistake.
Point 2: Original comment from reviewer 1 is that “How was this new method (RemL-C assay) applied to a population which was screened between 2012 and 2014?”
Reply from authors is that “Thank you for your comment on the RemL-C assay. The RemL-C assay kit was purchased from Minaris Medical Co. ltd. and used for the subjects. We rewrote some parts of the contents in the “Materials and Methods” section of the revised manuscript (Page 3, lines 120-122).”
Second comment from reviewer 1 is that “That means that the The RemL-C assay kit already exist in 2012? Is it really new?”
Response 2: As you mentioned, that is not so new methods. To avoid misleading representations, we rewrote a phrase in the “Introduction” section of re-revised manuscript.
Original (revised) (Page 2, line 75)
“More recently,”
Re-revised (Page 2, line 75)
“After the clinical application of RLP-C assay,”
Point 3: Original comment from reviewer 1 is that “In contrast to the RLP-C assay, the RemL-C assay can be performed on universal automated analyzers, which enables rapid, high-throughput measurements. Is it the unique advantage of this technology? If yes, I do not consider that it is a major progress.”
Reply from authors is that “Thank you for your suggestion. This method can be measured with an automated analyzer, does not require sample pretreatment, greatly shortens the measurement time (10 minutes), allows simultaneous measurement of multiple samples, reduces measurement errors, and improves measurement accuracy. In that sense, the RemL-C assay would be a more convenient method than the RLP-C assay. However, since it depends on the reader's point of view, we have not mentioned that in this article. To deepen the reader’s understanding of RemL-C assay, we added some characteristics of RemL-C assay in the “Discussion” section of the revised manuscript (Page 12, lines 324-327).”
Second comment from reviewer 1 is that “I still not have the same point of view.”
Response 3: Thank you for your suggestion. As you mentioned, the convenience of RemL-C assay is not general, and that depends on personal subjectivity. Our sentenses were biased toward RemL-C, so we deleted and rewrote some sentences to a more neutral expression as follow;
Original (revised) (Page 2, line 79-81)
“In contrast to the RLP-C assay, the RemL-C assay can be performed on a universal au-toanalyzer, which permits rapid, high-throughput measurements. Moreover,”
Re-revised (Page 2, line 79-80)
“In contrast to the RLP-C assay, the RemL-C assay can be performed on a universal au-toanalyzer, which permits rapid, high-throughput measurements. Moreover,” is deleted.
Original (revised) (Page 12, line 322-327)
“However, this measurement is relatively time-consuming and cannot be performed on an autoanalyzer. In contrast, the second method, the RemL-C assay, can be performed on a universal autoanalyzer, does not require sample pre-treatment, greatly shortens the measurement time (to 10 minutes), permits the simultaneous measurement of multiple samples, reduces measurement errors, and improves measurement accuracy.”
Re-revised (Page 12, line 338-340)
“The second method, the RemL-C assay, can be performed by homogeneous method [17]. Currently, these two measurements are selected by each investigator.”
Point 4: Original comment from reviewer 1 is that “In the discussion section, the authors explained that this is the first study to investigate the relationships between RemL-C and clinical parameters related to CVD risk in patients with T2DM. I did not find any clinical parameter in this study, like dyspnea classification, heart failure, or other clinical complications of T2DM.”
Reply from authors is that “Thank you for your suggestion. Unfortunately, as pointed out by the reviewers, other factors related to CVD such as dyspnea classification and heart failure are missing. Regarding this point, I added it to the limitation in the revised manuscript (Page 14, lines 433-435).”
Revised manuscript (Page 14, lines 433-435)
Discussion
“Third, in the present study, other factors related to CVD, such as dyspnea classification and the heart failure status, were not available. Further studies should address these issues in the future.”
Second comment from reviewer 1 is that “ I think that it is a major limitation of this study.”
Response 4: Thank you for your suggestion. According to your suggestion, we added a sentense in the re-revised manuscript.
Original (revised) (Page 14, line 433-435)
“Third, in the present study, other factors related to CVD, such as dyspnea classification and the heart failure status, were not available. Further studies should address these issues in the future.”
Re-revised (Page 14, line 446-448)
“Third, in the present study, other factors related to CVD, such as dyspnea classification and the heart failure status, were not available. It is a major limitation of the study. Further studies should address these issues in the future.”
Point 5: Regarding the reviewer 1’s comments, we received additional comments from two academic editors.
The academic editor 1’s comment is that “I think this report needs additional effort to better describe their findings according to reviewer 1. Especially, to clarify the concept of “new”, “clinical parameters” and the “overall conclusions”.
The Academic Editor 2’s comment is that “I agree with the opinion that further efforts are needed to better explain the findings of Reviewer 1. Therefore, in this paper, it is necessary to 1) describe “novelty” more clearly and politely, 2) describe “clinical parameters” in more detail, and 3) “overall conclusions” based on this finding. I would like to ask the authors to reconsider the careful and accurate description of the conclusions and findings based on this, and to revise them as major revision.”
Therefore, according to the suggestions from academic editors, we rewrote the several sentences in the Abstract, Results, and Conclusion section of re-revised manuscript as follow;
#Abstract” section
Original (Revised) (Page 1, lines 28-30)
“Thus, RemL-C may represent a useful index of RL particles in patients with T2DM, and further therapeutic interventions may be needed in the future, given that RemL-C represents a residual risk.”
Re-Revised manuscript (Page 1, lines 28-30)
“Thus, RemL-C may represent a useful index of lipid and glucose metabolism, and that may be a marker of peripheral atherosclerotic disease (PAD) in male patients with T2DM.”
“Results” section
3.1. Characteristics of the participants
Original (Revised) (Page 4, lines 179-181)
“Compared with participants with low RemL-C, those with higher RemL-C had higher FPG, HOMA-R, TC, TG, sdLDL-C, and ACR; and lower HDL-C, adiponectin, and ABI (Table 1).”
Re-Revised manuscript (from Page 4, line 178 to page 5, line 184)
“Compared with participants with low RemL-C, those with higher RemL-C had higher TC, TG and sdLDL-C, and lower HDL-C (Table 1). Besides lipid metabolism, higher RemL-C had higher FPG and HOMA-R, and lower adiponectin, suggesting that RemL-C was affected by obese-related glucose metabolism (Table 1). Interestingly, higher RemL-C had higher ACR and lower ABI, suggesting the relevance between RemL-C and diabetic nephropathy and/or peripheral arterial disease in T2DM.”
3.2. Relationships between RemL-C and clinical parameters in the participants with T2DM
Original (Revised) (Page 6, lines 200-203)
“Next, we investigated the relationships between RemL-C and clinical parameters in the participants with T2DM. Univariate analysis revealed that RemL-C positively correlated with FPG, FPI, HOMA-R, HbA1c, TC, TG, LDL-C, and sdLDL-C; and nega-tively correlated with HDL-C and adiponectin in participants with T2DM (Table 2).”
Re-Revised manuscript (Page 6, lines 203-209)
“Next, we investigated the relationships between a continuous variable of RemL-C and clinical parameters in the participants with T2DM using Pearson’s correlation coefficients. In relation to lipid metabolism, univariate analysis revealed that RemL-C positively correlated with TC, TG, LDL-C, and sdLDL-C, and negatively correlated with HDL-C (Table 2). Regarding glucose metabolism, RemL-C positively correlated with FPG, FPI, HOMA-R and HbA1c, and negatively correlated with adiponectin in participants with T2DM (Table 2).”
Original (Revised) (Page 7, lines 218-222)
“Univariate analysis revealed that RemL-C positively correlated with Apo-B, Apo-CII, Apo-CIII, and Apo E; and negatively correlated with Apo-AI (Table 3). Multivariate stepwise regression analysis revealed that Apo-AI, Apo-B, Apo-CIII, and Apo-E were significantly and independently associated with RemL-C (Table 3).”
Re-Revised manuscript (Page 7, lines 224-230)
“Univariate analysis revealed that RemL-C positively correlated with Apo-B, Apo-CII, Apo-CIII, and Apo-E, which were the major apolipoproteins in remnant lipoprotein particles; and negatively correlated with Apo-AI (Table 3). Multivariate stepwise regression analysis revealed that Apo-AI, Apo-B, Apo-CIII, and Apo-E were significantly and independently associated with RemL-C (Table 3), suggesting that RemL-C measurements reflected the serum concentrations of remnant lipoproteins.”
3.3. Relationships between high RemL-C and clinical parameters in T2DM
Original (Revised) (Page 7, lines 229-234)
“Next, to evaluate the relationships between high RemL-C and clinical parameters, lo-gistic regression analysis was performed in the participants with T2DM. Univariate analysis revealed that FPG, HOMA-R, TC, TG, HDL-C, LDL-C, sdLDL-C, ACR, adi-ponectin, and ABI were associated with higher concentrations (≥ 0.24 mmol/l) of RemL-C (Table 4). Multivariate logistic regression analysis showed that sdLDL-C and ABI were significantly and independently associated with high RemL-C concentration (Table 4).”
Re-Revised manuscript (Page 7, lines 237-242)
“Next, to evaluate the independent risk factors associated with high RemL-C, multivar-iate logistic regression analysis was performed in the participants with T2DM. For multivariate analysis, HOMA-R, sdLDL-C, eGFR, ACR, adiponectin, and ABI were se-lected as relevant factors from the results of univariate analysis (Table 1). As a result, sdLDL-C and ABI were significantly and independently associated with high RemL-C concentration (Table 4).”
3.4. Effect of sex on the relationships between RemL-C and various clinical parameters
Original (Revised) (Page 8, lines 249-254)
“Univariate analysis revealed that HOMA-R, TC, TG, HDL-C, LDL-C, sdLDL-C, and ABI were associated with higher RemL-C concentration in the male participants with T2DM (Table 5). Multivariate logistic regression analysis showed that sdLDL-C and ABI were significantly and independently associated with high RemL-C concentration in the men, as well as in the entire group of participants with T2DM (Table 5).”
Re-Revised manuscript (Page 8, lines 256-262)
“In relation to lipid metabolism, univariate analysis revealed that RemL-C positively correlated with TC, TG, LDL-C, and sdLDL-C, and negatively correlated with HDL-C (Table 5). Besides lipid metabolism, higher RemL-C had higher HOMA-R, and lower ABI (Table 5). For multivariate logistic regression analysis, HOMA-R, sdLDL-C, age, adiponectin, and ABI were selected as relevant factors from the results of univariate analysis (Table 5). As a result, sdLDL-C and ABI were significantly and independently associated with high RemL-C concentration in the men, as well as in the entire group of participants with T2DM (Table 5).”
Original (Revised) (Page 8, lines 254-258)
“Interestingly, in the female participants, univariate analysis revealed that age, FPG, TG, HDL-C, LDL-C, sdLDL-C, and adiponectin were associated with high RemL-C concen-tration, and multivariate logistic regression analysis showed that sdLDL-C alone was significantly and independently associated with high RemL-C concentration (Table 6).”
Re-Revised manuscript (Page 8, lines 264-269)
“Interestingly, in the female participants, univariate analysis revealed that age, FPG, TG, HDL-C, LDL-C, sdLDL-C, and adiponectin were associated with high RemL-C concentration (Table 6). For multivariate logistic regression analysis, ABI, HOMA-R, sdLDL-C, adiponectin, and age were selected as relevant factors from the results of univariate analysis (Table 6). As a result, sdLDL-C alone was significantly and independently associated with high RemL-C concentration in the women (Table 6).”
3.5. Differences in several parameters between participants with or without a history of CVD in T2DM
Original (Revised) (Page 9, lines 292-297)
“Finally, the differences in parameters in participants with T2DM who did or did not have a history of CVD were evaluated. Age, the duration of T2DM, FPI, Lp(a), mean CCA-IMT, and max CCA-IMT were significantly higher in participants with CVD than in those without CVD (Table 7), whereas TC, LDL-C, eGFR, and ABI were significantly lower (Table 7). There were no differences in RemL-C, sdLDL-C, HDL-C, TG, SBP, DBP, BMI, or HOMA-R between these groups (Table 7).”
Re-Revised manuscript (Page 9, lines 303-313)
“Finally, the differences in parameters in participants with T2DM who did or did not have a history of CVD were evaluated. Age, the duration of T2DM, FPI, Lp(a), mean CCA-IMT, and max CCA-IMT, which were thought to be risk factors for CVD, were significantly higher in participants with CVD than in those without CVD (Table 7). Moreover, eGFR, which was a marker of chronic kidney disease, and ABI, which was a marker of peripheral atherosclerotic disease (PAD) were significantly lower in participants with CVD (Table 7). However, TC and LDL-C, which were strongly risk factors for CVD, were oppositely and significantly lower in participants with CVD (Table 7), suggesting the effect of statin use. Moreover, there were no differences in other risk factors for CVD, such as RemL-C, sdLDL-C, HDL-C, TG, SBP, DBP, BMI, or HOMA-R between these groups (Table 7), suggesting the existence of residual risk for CVD.”
“Conclusion” section
Original (Revised) (Page 14, lines 437-444)
“In the present study, we have demonstrated that high RemL-C concentration is associated with several factors related to CVD in patients with T2DM. Moreover, logistic regression analysis revealed that sdLDL-C and ABI are independently associated with a high RemL-C concentration. Moreover, there was no difference in the RemL-C concentration between individuals with or without a history of CVD. Therefore, RemL-C concentration may be a useful index of remnant lipoproteins, and further therapeutic interventions targeting RemL-C may be needed in the future, given the residual risk associated with RemL-C.”
Re-Revised manuscript (Page 14, lines 450-459)
“In the present study, we have demonstrated in T2DM that i) RemL-C is associated with apolipoproteins related with remnant lipoproteins, ii) high RemL-C concentration is associated with several factors related to lipid and glucose metabolism, iii) sdLDL-C are independently associated with a high RemL-C concentration in both female and male subjects, iv) ABI are independently associated with a high RemL-C concentration in male subjects, and v) there was no difference in the RemL-C concentration between individuals with or without a history of CVD, possibly due to statin use. Therefore, RemL-C concentration may be a useful index of lipid and glucose metabolism, and that may be a marker of PAD in male subjects with T2DM. Regarding RemL-C as a risk factor for CVD, further prospective observational studies are needed in the future.”
We would like to express our sincere appreciation to you for your very helpful suggestions and comments. We think our re-revised version is now clearer and more scientific.
Thank you very much again.

Reviewer 3 Report
I received a revised version of the paper for re-evaluation. I believe that the article in its current version is very well prepared. It represents high scientific and cognitive value. The research methodology and the results obtained were precisely described and a thorough discussion was conducted. I recommend the article for publication in its current version.
Author Response
Thank you for your kind words. Your advice made the manuscript more scientific. Please allow me to thank you again.